# SwiftMax: Reducing Training Time for Learnable Softmax Alternative in Customized Acceleration

## Abstract

Softmax's row-wise max and sum impose an $O(n)$ normalizer substep inside self-attention, creating latency and bandwidth bottlenecks on modern accelerators. We introduce **SwiftMax**, a drop-in, learnable alternative that replaces these reductions with per-layer scalars $\beta, \gamma$, removing the length-$n$ dependency in the normalizer while leaving $QK^{\top}$ and value mixing unchanged. SwiftMax is enabled by a *layer-wise replace-and-tune* schedule that updates only $\beta, \gamma$ on top of a frozen pretrained model; initialization is guided by the output statistics of the Softmax normalizer (distributions of $z_{\max}$ and $\sum_j e^{z_j - z_{\max}}$). On BERT-base across GLUE, SwiftMax matches the Softmax baseline within 1–3 accuracy points on SST-2/MNLI/QQP, with a larger drop on RTE; compared with approaches that retrain all parameters to learn these scalars (e.g., ConSmax-style training), SwiftMax cuts end-to-end training time by orders of magnitude (up to 2,250× in our setting). On AMD ACAP, eliminating the row dependency enables up to 23× speedup for the self-attention normalizer and substantial module-level gains, alleviating pipeline stalls and memory traffic. Taken together, SwiftMax offers a practical path to hardware-friendly attention with minimal accuracy loss and without full retraining, bridging the gap between pretrained models and custom acceleration.

## 1 Introduction

Transformer models (Vaswani et al., 2017) have become the cornerstone of advancements in natural language processing (NLP) and computer vision (CV) due to their ability to capture long-range dependencies through Self-Attention mechanism. A pivotal component of these mechanisms is the Softmax function. However, as sequence lengths increase, the Softmax operation becomes a significant bottleneck.

To address these challenges, several methods have been proposed to approximate or replace the Softmax function to reduce computational overhead. Various partial Softmax implementations, spearheaded by FlashAttention (Dao et al., 2022), aim to optimize the computation by partitioning the operation, but they retain the row-wise dependencies and the $O(n)$ per-row reduction cost of the *normalizer substep*, limiting parallelism and memory efficiency. Softermax (Stevens et al., 2021) simplifies the computation but does not remove the length-$n$ reductions. ConSmax (Liu et al., 2024) replaces the maximum and summation reductions with learnable

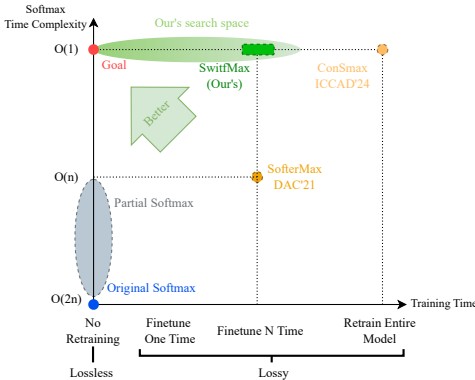

Figure 1: Comparison Among Different Softmax Alternatives. Goal: minimize retraining time while achieving $O(1)$ per-row complexity *for the Softmax normalizer substep* (max + sum) via learned constants.

parameters, eliminating the per-row reduction cost of the normalizer, but it requires retraining the entire model. To overcome these limitations, we propose **SwiftMax**, a learnable Softmax alternative

designed to minimize training time and accuracy loss while alleviating the computational bottleneck in Self-Attention mechanism. Our key contributions are as follows:

- **Analyzing Softmax Output Statistics in Pre-trained Models**: We analyze the output statistics (distribution of $z_{\max}$ and the normalizer $\sum_j e^{z_j - z_{\max}}$) in the Self-Attention layers of pre-trained models. Based on these statistics, we replace Softmax normalizers with SwiftMax and fine-tune only the introduced scalars, leveraging existing knowledge without full retraining.

- **Reducing Training Time by Fine-Tuning**: By merging the concepts of efficient fine-tuning and seamless integration with pre-trained models, our method eliminates the need for retraining the entire model. Instead, we fine-tune only the newly introduced parameters. This significantly reduces computational resources and time—our approach saves up to $2,250\times$ in training time compared to full retraining methods like ConSmax, while maintaining over 80% of the original model's accuracy in most tasks.

- **Deploying SwiftMax on ACAP Platform**: AMD Adaptive Compute Acceleration Platforms (ACAP) provides flexible customization strategies for computation and data flow. Based on ACAP, the deployment of SwiftMax achieves a balance between inference speed and accuracy, achieving up to $23\times$ performance improvement during inference. Our experiments demonstrate that SwiftMax effectively alleviates the Softmax bottleneck in Self-Attention mechanism, enabling efficient deployment of Transformer models in hardware-constrained environments.

## 2 PRELIMINARIES AND RELATED WORK

### 2.1 SOFTMAX BOTTLENECKS

The Softmax function is defined as:

$$\text{Softmax}(z_i) = \frac{e^{z_i - z_{\max}}}{\sum_{j=1}^{n} e^{z_j - z_{\max}}} \tag{1}$$

Calculating $z_{\max}$ for numerical stability and then the denominator summation requires two sequential length-$n$ reductions (max + sum) per row, i.e., $O(n)$ work with unavoidable row-wise dependency for the *normalizer substep*. This substep—while not dominating the full $O(n^2 d)$ cost of attention—can become a latency and memory bottleneck in hardware pipelines.

- **Limited Parallelism:** The inherent sequential dependency in computing the Softmax function limits parallelization. Each output depends on all elements in the input vector, making it difficult to parallelize computations as efficiently as matrix multiplications.

- **Memory Bandwidth Bottlenecks and Usage Issues:** The need to access the entire input vector for normalization leads to high memory bandwidth demands. Additionally, storing intermediate exponential values increases memory usage, which is problematic for deploying large models on hardware with limited resources.

### 2.2 SOFTMAX ACCELERATION

**Lossless Softmax Alternatives.** Various methods aim to preserve the normalization properties of the Softmax function while improving computational efficiency. FlashAttention (Dao et al., 2022) represents a partial Softmax solution that segments the computation to enhance data locality but retains the per-row reduction dependency. Online Softmax (Milakov & Gimelsheins, 2018) restructures the reductions but does not eliminate them. **Lossy Softmax Alternatives.** Several works have focused on relaxing Softmax's strict functional behavior to achieve substantial speedups.

Softermax (Stevens et al., 2021) simplifies the computation using base-2 exponentiation to reduce power consumption and computational demands. CosFormer (Qin et al., 2022) proposes a linear transformer that substitutes the Softmax attention with a cosine-based re-weighting mechanism, maintaining non-negativity and distribution concentration while achieving competitive accuracy and

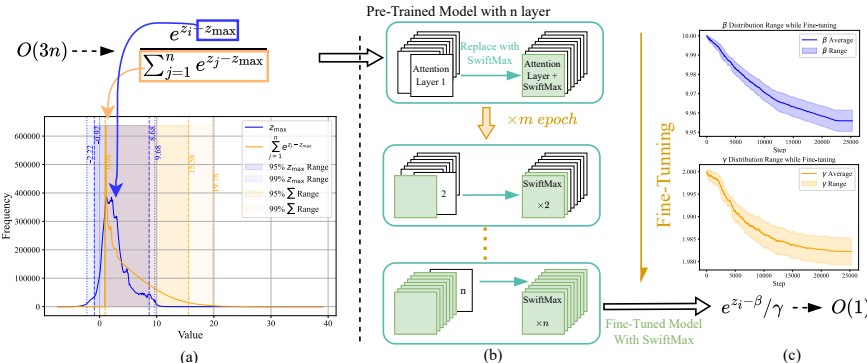

(a)  (b)  (c)

Figure 2: SwiftMax Workflow: (a) Softmax output statistics range ($z_{\max}$ and normalizer). (b) Layer-wise replacement process. (c) SwiftMax parameter learning.

strong performance on long-sequence benchmarks. ConSmax (Liu et al., 2024) removes the max and sum reductions by learning scalars, thus yielding $O(1)$ cost for the normalizer substep (not the entire attention), but requires end-to-end retraining.

## 2.3 ACCELERATION ON ACAP

Many works have been developed to build accelerators based on ACAP (Vissers, 2019; Ahmad et al., 2019). CHARM (Zhuang et al., 2023) is the pioneering work in the domain of ACAP Transformer acceleration, providing matrix multiplication operators and enabling automated code generation. SSR (Zhuang et al., 2024), the follow-up work (currently SOTA), constructs accelerators by employing a spatially sequential mixed scheduling strategy for computation units. The EA4RCA framework(Zhang et al., 2024b) utilizes the CA algorithm to maximize the utilization of AIE for a single application. CAT(Zhang et al., 2024a) focuses on AIE, deploying the entire Transformer Layer onto hardware as a unified unit. $G^2PM$ (Dai et al., 2024) provides comprehensive performance modeling for the ACAP architecture.

## 3 PROPOSED ALGORITHM

### 3.1 THEORETICAL FOUNDATION

From the above, it is evident that to achieve maximum performance, methods like ConSmax must be employed to address the row-wise dependency issue of the Softmax. By eliminating the row-wise reduction dependency inside the Softmax normalizer (max + sum), the per-row normalizer substep cost changes from two length-$n$ reductions to constant-time scalar operations, improving pipeline parallelism; the overall attention still requires $QK^\top$ and value mixing with $O(n^2 d)$ complexity.

ConSmax has revealed that $z_{\max}$ and $\sum_{j=1}^{n} e^{z_j - z_{\max}}$ can be replaced by learnable parameters $\beta$ and $\gamma$. These parameters can be optimized through training the entire model to achieve numerically stable values. The ConSmax is defined as:

$$\text{ConSmax}(z_i) = \frac{e^{z_i - \beta}}{\gamma} \tag{2}$$

The ConSmax retains the differentiability of backpropagation in neural networks. This ensures that the model can effectively use gradient-based optimization methods for fine-tuning, and SwiftMax can seamlessly integrate into existing pre-trained models without compromising training stability.

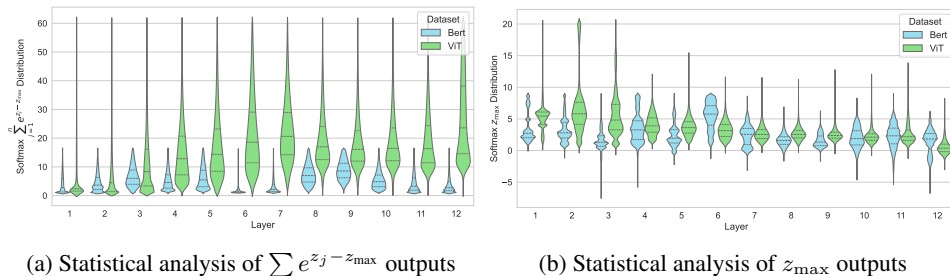

(a) Statistical analysis of $\sum e^{z_j - z_{\max}}$ outputs       (b) Statistical analysis of $z_{\max}$ outputs

Figure 3: Statistical Analysis of Softmax Outputs

However, the main limitation of the ConSmax lies in its requirement to retrain the entire model to learn the parameters $\beta$ and $\gamma$. This retraining process demands extensive computational resources and access to the original training data, which is neither practical nor feasible for most users.

SwiftMax is built upon the insight that these scalars can be estimated from Softmax output statistics of pre-trained models and refined via lightweight fine-tuning instead of full retraining. By deeply analyzing the behavior of the Softmax and its role in the model, we developed strategies to estimate appropriate parameters, ensuring that the original Softmax's functionality is approximated during inference.

Thus, our work focuses on identifying suitable ConSmax parameters for pre-trained models without retraining, thereby achieving performance improvements.

To achieve this goal, we require a method that maintains model accuracy, significantly improves computational efficiency, and avoids retraining the entire model. To this end, we considered the following key factors:

- **Utilizing Information from Pre-Trained Models**: We aim to leverage the existing information in pre-trained models to extract or constrain $\beta$ and $\gamma$, maximizing the performance of existing models.
- **Maintaining Model Stability and Performance**: During the replacement of the Softmax, as ConSmax is no longer strictly normalized, it is essential to ensure numerical stability and prevent significant degradation in accuracy.
- **Reducing Training Time**: We seek an approach that avoids large-scale modifications to the model structure or training process, making the replacement process simple, practical, and deployable.

## 3.2 STRATEGY SELECTION

Based on the considerations above, we explored methods to obtain suitable fixed parameters $\beta$ and $\gamma$ under different training loads. The goal is to find the most effective solution that balances model performance and training time while maintaining accuracy.

Initially, we attempted statistical analysis of the Softmax outputs from fine-tuned models as illustrated in Fig. 3. We collected the Softmax outputs from each layer and attention head, calculating the corresponding $z_{\max}$ and $\sum_{j=1}^{n} e^{z_j - z_{\max}}$ values. The results indicated that these values exhibited a distribution similar to a normal distribution. Based on this observation, we tried using the statistical averages of these values as fixed parameters $\beta$ and $\gamma$, aiming to approximate the original Softmax.

However, due to the diversity of input data and the complexity of internal model structures, fixed statistical averages or percentiles could not adapt to all scenarios. This led to numerical instability during inference and severe overflow issues, rendering the model unusable. Thus, the strategy of directly using statistical values as fixed parameters did not achieve the expected results.

To better adapt to variations in input data and maintain numerical stability, we introduced fine-tuning to adjust the fixed parameters within the model. Specifically, $\beta$ and $\gamma$ were treated as learnable parameters, allowing the model to adjust their values during fine-tuning. Based on this, we explored two strategies:

- **One-Time Replacement and Fine-Tuning:** We attempted to replace all Softmax in the model with ConSmax with learnable parameters at once, followed by fine-tuning the entire model to learn optimal $\beta$ and $\gamma$. However, this abrupt global replacement made it difficult for the model to adapt to the new activation function, leading to unstable training and convergence issues.

- **Layer-Wise Replacement and Fine-Tuning:** From shallow to deep layers, we progressively replaced the Softmax function in each layer with the ConSmax function. After replacing each layer, we fine-tuned the model to adapt it gradually to the new activation function. This progressive replacement method effectively improved training stability, enabling the model to successfully learn suitable $\beta$ and $\gamma$ values while maintaining numerical stability.

### 3.3 SWIFTMAX ALGORITHM

In SwiftMax, $\beta$ and $\gamma$ are introduced as learnable parameters, replacing the maximum value and normalization factor in Softmax, respectively. To effectively apply SwiftMax to a pre-trained model, we designed a layer-wise replacement and fine-tuning algorithm. The detailed steps are as follows:

---

**Algorithm 1** Progressive Layer Replacement with Fine-Tuning

---

**Require:** Pre-trained Transformer model with $N$ layers
**Require:** Number of epochs per stage $E$
**Require:** Initial SwiftMax parameters $\beta_l$ and $\gamma_l$
1: **for** $l = 1$ to $N$ **do**
2:    Replace the Softmax function in layer $l$ with **SwiftMax**
3:    Initialize $\beta_l$ and $\gamma_l$ in SwiftMax as learnable parameters
4:    **for** epoch $= 1$ to $E$ **do**
5:       **for** mini-batch $b$ in training set **do**
6:          Compute model output with current parameters
7:          Compute loss $\mathcal{L}$ on mini-batch $b$
8:          Backpropagate gradients
9:          Update $\beta_l$ and $\gamma_l$
10:       **end for**
11:       Evaluate model on validation set
12:    **end for**
13: **end for**

---

Algorithm 1 provides a detailed description of the process for progressively replacing Softmax with SwiftMax and fine-tuning. In each layer, we fine-tune the newly replaced learnable parameters, $\beta_l$ and $\gamma_l$, along with the main model parameters. This gradual approach allows the model to adapt incrementally to the new activation function, reducing the risk of instability during training.

This method introduces only a small number of modifications each time, enabling a more precise identification of appropriate values for $\beta$ and $\gamma$ while maintaining overall performance. This approach achieves an effective balance between implementation complexity and model performance.

In the above algorithm, selecting key hyperparameters has a significant impact on both the model's performance and training time. Through experimentation, we determined the following optimal hyperparameter settings:

- **Epochs per Stage** ($E$): the number of fine-tuning epochs performed after each replacement. A small $E$ (e.g., 2–3) is sufficient for adaptation and prevents unnecessarily long training.

- **Layers per Stage** ($L$): the number of layers whose normalizers are replaced in one stage (default $L = 1$ in Algorithm 1). Larger $L$ accelerates replacement but may destabilize training.

- **Parameter Initialization**: $\beta$ and $\gamma$ are initialized with the statistical means of $z_{\max}$ and the normalizer in the pre-trained model. To ensure numerical stability, $\gamma$ is initialized positive.

- **Learning Rate**: the learning rate $\eta$ controls the convergence of both frozen and newly introduced scalars. In practice, we freeze the main weights (or use a small $\eta$) and apply a moderately larger $\eta$ to $\beta, \gamma$ for faster convergence without overshooting.

With these hyperparameter configurations, the SwiftMax algorithm can significantly reduce training time while ensuring model accuracy in most tasks, achieving an efficient adaptation to pre-trained models. The impact of each hyperparameter on SwiftMax performance will be evaluated in detail in the Section 4.2.

### 3.4 MINIMIZING SWIFTMAX DEPLOYMENT ON ACAP

We implemented a customizable Self-Attention Block (ATB) on ACAP, enabling the dynamic replacement of Softmax to SwiftMax. To achieve efficient Self-Attention mechanism, we deployed a computation engine on the AIE for handling intensive computational tasks and designed a data engine on the PL side to support the AIE. The two components work in a pipelined parallel manner to ensure high-speed data flow. To dynamically switch between SwiftMax and Softmax, we adopted a dataflow reconstruction strategy and hardware resource reuse strategy to enable replacement and reorganization within the ATB module. Moreover, since SwiftMax eliminates intra-row dependencies, we integrated SwiftMax into the data engine, achieving even higher performance.

## 4 EXPERIMENT

### 4.1 EXPERIMENTAL SETUP

To evaluate the effectiveness of SwiftMax, we conducted experiments on two widely used models: the **BERT-base** model (Devlin et al., 2019) in the field of NLP and the **ViT-base** model (Dosovitskiy et al., 2021) in CV. The corresponding datasets are the **GLUE** benchmark (Wang et al., 2019) and the **CIFAR-10** dataset (Krizhevsky & Hinton, 2009).

During implementation, we utilized a modified HuggingFace PyTorch library (Paszke et al., 2019) to inject SwiftMax operations into the model, replacing the Softmax function in the Self-Attention mechanism. By extending HuggingFace's modular design, we were able to flexibly apply SwiftMax without altering the overall structure of the model.

The experiments adopted a layer-wise replacement and fine-tuning strategy, as shown in Algorithm 1. After replacing Softmax with SwiftMax in each layer, we fine-tuned the model to gradually adapt to the new activation function, reducing training instability.

All experiments were conducted in a computing environment equipped with NVIDIA RTX4090 GPU. Our experimental code was implemented based on the PyTorch and HuggingFace Transformers libraries.

### 4.2 HYPERPARAMETER TUNING AND ANALYSIS

In this section, we discuss in detail the impact of key hyperparameters on model performance and present the corresponding experimental results. We mainly focus on the following three key factors: **Epochs per Stage** $E$: the number of layers replaced each time $E$ affects the model's training stability and total training time. **Initialization of SwiftMax Parameters**:

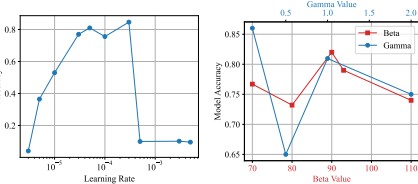

(a) Effect of Learning Rate $\eta$ on Validation Accuracy

(b) Impact of SwiftMax Parameter Initialization on Validation Accuracy

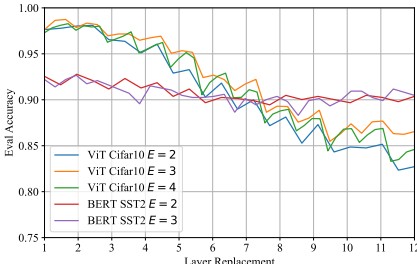

(c) Effect of Epochs per Stage $E$ on Model Performance

Figure 4: Impact of Key Hyperparameters on Model Performance

the initial values of $\beta$ and $\gamma$ affect the convergence

speed of parameters and the numerical stability of the model. **Choice of Learning Rates**: the learning rate $\eta$ for main model parameters.

As shown in Figure 4a, the choice of learning rate has a significant impact on the model's convergence speed and final performance. In our subsequent experiments, we used a learning rate of $3 \times 10^{-5}$ for both BERT and ViT models, utilizing the AdamW optimizer (Loshchilov & Hutter, 2019) with an L2 weight decay of 0.01. We found that this setting for the main model parameters' learning rate $\eta$ helps to achieve a balance between convergence speed and avoiding overfitting.

In Figure 4b, we explore the impact of the initial values of SwiftMax parameters $\beta$ and $\gamma$ on model training. Appropriate initial values help the parameters quickly converge to a reasonable range, ensuring the numerical stability of the model. Through experiments, we found that initializing $\beta$ and $\gamma$ with the statistical average values of the corresponding parameters in the pre-trained model yields good results.

Figure 4c illustrates the impact of the number of layers replaced per stage on model performance and training time. We observed that for the BERT model, fine-tuning for 2 epochs after each replacement was sufficient to achieve the desired performance. However, for the ViT model, the number of fine-tuning epochs had a more pronounced effect on the model's performance. To balance accuracy and prevent overfitting, we chose to fine-tune for 3 epochs after each replacement for the ViT model. This adjustment allowed us to maintain high accuracy while avoiding the pitfalls of overfitting, highlighting the importance of tailoring the fine-tuning process to the specific characteristics of each model.

### 4.3 EVALUATION

Based on the optimal hyperparameter combinations found in Section 4.2, we conducted a comprehensive evaluation of the SwiftMax strategy by applying it to the BERT-base and ViT-base models. Using the layer-wise replacement and fine-tuning strategy described in Algorithm 1, we integrated SwiftMax into the models and evaluated their performance on their respective datasets to assess the impact on model accuracy and training time.

Table 1: BERT-base on GLUE Benchmark

| Model | SST-2 | MNLI-(m/mm) | QQP | RTE |
|---|---|---|---|---|
| BERT$_{BASE}$ | 93.5 | 84.6/83.4 | 71.2 | 66.4 |
| BERT$_{BASE}$ + SwiftMax | 92.6 | 83.0/81.5 | 69.5 | 52.2 |
| Performance Loss | 1.0% | 1.9%/2.4% | 2.4% | 21.4% |

For the BERT model on the GLUE benchmark, adopting SwiftMax resulted in only a slight decrease in accuracy across most tasks. This minimal performance drop demonstrates that our layer-wise replacement and fine-tuning strategy effectively maintains model performance while significantly reducing training time and computational complexity. The results confirm that SwiftMax is well-suited for NLP tasks, where the Softmax parameter distribution is relatively narrow, allowing for efficient convergence during fine-tuning.

Compared to the substantial computational resources required to train the entire BERT model, which takes 384 TPU-hours, our method requires only a fraction of that time, from as little as 10 minutes for the RTE task to 5.5 hours for the QQP task on a single RTX4090 GPU. This results in a speedup of up to $2,250\times$, significantly reducing the computational resources and time required for training. This dramatic reduction in training time makes our method highly practical for rapid deployment and experimentation.

In contrast, the ViT model on the CIFAR datasets experienced a more pronounced decrease in performance, especially on CIFAR-100. From Fig. 3, we observe that the distribution of Softmax parameters in ViT is much wider than in BERT. This wider distribution leads to greater difficulty in convergence during fine-tuning, resulting in more significant performance degradation. Despite the substantial accuracy loss, this behavior highlights an important characteristic of ViT models: their sensitivity to changes in the attention mechanism due to the complex and high-dimensional nature of visual data.

To address the performance loss in the ViT model, we can leverage the flexibility of customizable accelerators during deployment. By replacing only a portion of the Self-Attention layers with Swift-Max, we can balance the trade-off between model accuracy and inference speed.

## 4.4 SYSTEM VERIFICATION

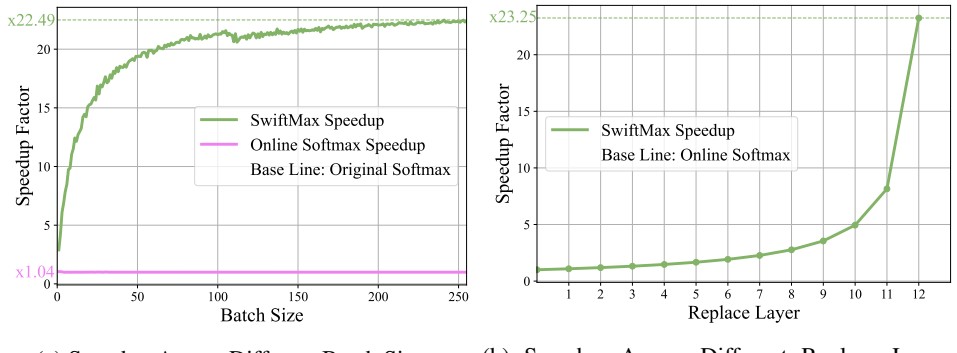

(a) Speedup Across Different Batch Sizes.

(b) Speedup Across Different Replace Layers (Batch Size = 256).

Figure 5: Performance improvement of the Self-Attention layer with SwiftMax compared to Softmax on VCK5000.

SwiftMax demonstrates significant performance improvements on the AMD ACAP platform. As shown in Fig. 5, SwiftMax achieves up to a $23\times$ speedup in the Self-Attention modules of BERT and ViT models compared to traditional Softmax functions and Online Softmax (Milakov & Gimelsheins, 2018) on the VCK5000 platform. The tests were conducted with different batch sizes to evaluate its scalability and performance under various workloads. This substantial acceleration is attributed to the elimination of row-wise dependencies and the reduction of computational complexity from $O(n)$ to $O(1)$, enabling highly parallel computation and full utilization of the hardware resources of the VCK5000.

Table 2: Resource Utilization Comparison between SwiftMax and Online Softmax in the ATB on AMD ACAP

| Softmax implementation | LUT | FF | LUTRAM | BRAM |
|---|---|---|---|---|
| Online Softmax | 115488 | 103197 | 3286 | 74 |
| SwiftMax | 38473 | 60424 | 5561 | 90 |
| Relative Resource Usage | 33% | 58% | 59% | 121% |

In terms of resource utilization on the AMD ACAP platform, SwiftMax demonstrates significant advantages over Online Softmax implementations. As shown in Table 2, SwiftMax requires far fewer of the Look-Up Tables (LUT) and the Flip-Flop (FF) compared to Online Softmax. While it uses slightly more Block RAM (BRAM), the efficient usage of resources allows for more complex models or additional functionalities to be deployed on the same hardware platform.

Furthermore, as illustrated in Fig. 5, even when SwiftMax is not fully utilized across all layers on the ACAP platform, it still provides a substantial performance improvement. This partial deployment allows for flexible trade-offs between inference speed and model accuracy, enabling practitioners to balance performance and precision according to specific application requirements.

On general-purpose GPU platforms, SwiftMax also achieves performance improvements. With a batch size of 32 and sequence length of 2048, we observed a $1.17\times$ speedup using PyTorch implementations. However, while this method is effective on GPU, its advantages are more pronounced with customized hardware like ACAP. SwiftMax is most prominent when computations are parallelized, blocked, and pipelined, fully leveraging the ACAP architecture's capabilities.

## 5 CONCLUSION

In this paper, we proposed SwiftMax, an efficient alternative to the Softmax function in Transformer models that eliminates the row-wise dependency of Softmax, reducing computational complexity from $O(n)$ to $O(1)$. Our layer-wise replacement and fine-tuning strategy enables SwiftMax to be seamlessly applied to pre-trained models without the need for full retraining, reducing training time by up to 2,250 times, while maintaining model accuracy. We achieved up to 23× performance improvement, effectively mitigating the Softmax performance bottleneck in Self-Attention mechanism. This approach provides a practical solution for efficiently deploying Transformer models in hardware-constrained environments.

In the future, we aim to further optimize the parameter learning methods of SwiftMax and explore its performance on larger-scale models and a wider range of application domains.

## REPRODUCIBILITY STATEMENT

We release minimal code and configs in the supplementary materials (App. B). Section 4.2 details all hyperparameters, initialization, and schedules. Datasets and preprocessing steps follow standard GLUE/CIFAR protocols with references to their official releases. We seed all runs and report exact environment versions in App. B.1.

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

## A    Use of Large Language Models (LLMs)

In accordance with the ICLR 2026 policy, we disclose the use of large language models (LLMs) during the preparation of this work. LLMs were employed in the following limited ways:

- **Writing assistance:** We used LLMs to polish grammar, improve clarity, and adjust the flow of the paper text. The core technical ideas, algorithm design, theoretical analysis, and experimental results were fully developed by the authors.
- **Code assistance:** LLMs were used to generate draft snippets of experimental code (e.g., Python scripts for layer replacement and fine-tuning). All generated code was carefully verified, adapted, and debugged by the authors before integration into experiments.

LLMs were not used for research ideation, data analysis, or interpretation of results. All scientific contributions, designs, and conclusions presented in this paper are the sole responsibility of the authors.

## B    Core Code (Minimal)

This appendix contains the minimal SwiftMax operator, the injection into `BertSelfAttention`, and the progressive layer-replacement callback used to realize Algorithm 1. We intentionally omit logging, visualization, and data collection utilities to keep the code concise and reproducible.

### B.1    Environment and Versions

```
GPU: NVIDIA RTX 4090
Python: 3.12.7
packages:
datasets==3.0.2
evaluate==0.4.3
h5py~=3.12.1
matplotlib==3.9.2
numpy>=2.0.2
optimum==1.23.3
pandas==2.2.3
torch==2.6.0
torchvision==0.20.1
transformers==4.46.0
scikit-learn==1.5.2
safetensors==0.4.5
accelerate>=0.26.0
tqdm~=4.66.6
seaborn~=0.13.2
scipy~=1.14.1
tensorboard~=2.18.0
```

**Scope.**    Unless otherwise noted, we train only the newly introduced SwiftMax parameters $\beta$ and $\gamma$ (normalizer scalars); all other model components remain architecturally unchanged.

### B.2    SwiftMax Operator (Core)

```python
# swiftmax_core.py
import math
import torch
from torch import nn

device = torch.device("cuda" if torch.cuda.is_available() else "cpu")

class SwiftMax(nn.Module):
    """
    SwiftMax computes: out = base**(x - beta) / gamma
    where beta, gamma are learnable scalars.
```

```python
    Clamping stabilizes training and avoids overflow under AMP.
    """
    def __init__(self, base: float = math.e, beta_init: float = 10.0,
    ↪   gamma_init: float = 2.0):
        super().__init__()
        self.base = float(base)
        self.beta  = nn.Parameter(torch.tensor([beta_init],
        ↪   device=device))
        self.gamma = nn.Parameter(torch.tensor([gamma_init],
        ↪   device=device))

    @torch.cuda.amp.autocast(enabled=False)
    def forward(self, x: torch.Tensor) -> torch.Tensor:
        x32 = x.to(dtype=torch.float32)
        x_adj = torch.clamp(x32 - self.beta, min=-10.0, max=40.0)
        e_x = torch.pow(torch.tensor(self.base, device=x.device,
        ↪   dtype=x32.dtype), x_adj)
        out = e_x / self.gamma
        return out.to(dtype=x.dtype)
```

**Notes on stability.** We clamp $(x-\beta)$ to $[-10, 40]$. The default base is $e$; using base 2 can simplify certain hardware realizations at the expense of a small distributional shift.

### B.3    MINIMAL INJECTION INTO BERTSELFATTENTION

```python
# bert_swiftmax_inject.py
import math
import torch
from transformers.models.bert.modeling_bert import BertSelfAttention
from swiftmax_core import SwiftMax

class SwiftMaxBertSelfAttention(BertSelfAttention):
    """
    Drop-in replacement: replaces softmax(attn_scores) with
    ↪   SwiftMax(attn_scores).
    """
    def __init__(self, config, position_embedding_type=None):
        super().__init__(config, position_embedding_type)
        base  = getattr(config, "swiftmax_base", math.e)
        beta0 = getattr(config, "swiftmax_initial_beta", 10.0)
        gamma0= getattr(config, "swiftmax_initial_gamma", 2.0)
        self.swiftmax = SwiftMax(base=base, beta_init=beta0,
        ↪   gamma_init=gamma0)

    def forward(
        self,
        hidden_states: torch.Tensor,
        attention_mask: torch.FloatTensor | None = None,
        head_mask: torch.FloatTensor | None = None,
        encoder_hidden_states: torch.FloatTensor | None = None,
        encoder_attention_mask: torch.FloatTensor | None = None,
        past_key_value=None,
        output_attentions: bool = False,
    ):
        mixed_query_layer = self.query(hidden_states)
        is_cross_attention = encoder_hidden_states is not None

        if is_cross_attention and past_key_value is not None:
            key_layer, value_layer = past_key_value
            attention_mask = encoder_attention_mask
        elif is_cross_attention:
            key_layer   =
            ↪   self.transpose_for_scores(self.key(encoder_hidden_states))
            value_layer =
            ↪   self.transpose_for_scores(self.value(encoder_hidden_states))
```

```
                    attention_mask = encoder_attention_mask
            elif past_key_value is not None:
                key_layer   =
                ↪    self.transpose_for_scores(self.key(hidden_states))
                value_layer =
                ↪    self.transpose_for_scores(self.value(hidden_states))
                key_layer   = torch.cat([past_key_value[0], key_layer],
                ↪    dim=2)
                value_layer = torch.cat([past_key_value[1], value_layer],
                ↪    dim=2)
            else:
                key_layer   =
                ↪    self.transpose_for_scores(self.key(hidden_states))
                value_layer =
                ↪    self.transpose_for_scores(self.value(hidden_states))

            query_layer = self.transpose_for_scores(mixed_query_layer)
            use_cache = past_key_value is not None
            if self.is_decoder:
                past_key_value = (key_layer, value_layer)

            attention_scores = torch.matmul(query_layer,
            ↪    key_layer.transpose(-1, -2))
            attention_scores = attention_scores /
            ↪    math.sqrt(self.attention_head_size)

            if attention_mask is not None:
                attention_scores = attention_scores + attention_mask

            # SwiftMax replaces the softmax normalizer
            attention_probs = self.swiftmax(attention_scores)
            if attention_mask is not None:
                masked = (attention_mask < -1e9)
                raw = raw.masked_fill(masked, 0.0)
            attention_probs = self.dropout(attention_probs)

            if head_mask is not None:
                attention_probs = attention_probs * head_mask

            context_layer = torch.matmul(attention_probs, value_layer)
            context_layer = context_layer.permute(0, 2, 1, 3).contiguous()
            new_shape = context_layer.size()[:-2] + (self.all_head_size,)
            context_layer = context_layer.view(new_shape)
            outputs = (context_layer, attention_probs) if output_attentions
            ↪    else (context_layer,)
            if self.is_decoder:
                outputs = outputs + (past_key_value,)
            return outputs
```

**Remark.** The projections (Q/K/V) and dropout remain exactly as in the original implementation; only the probability normalizer is modified.

## B.4 PROGRESSIVE REPLACEMENT CALLBACK (BERT)

```
# progressive_callback.py
import math
from transformers import TrainerCallback
from bert_swiftmax_inject import SwiftMaxBertSelfAttention

class ReplaceBertLayersCallback(TrainerCallback):
    """
    Every `epochs_per_stage` epochs, replace the next `layers_per_stage`
    self-attention modules with SwiftMax. Newly introduced beta/gamma are
    added to the main optimizer's first param group.
    """
```

```python
    def __init__(self, model, layers_per_stage: int, epochs_per_stage:
      int, swiftmax_lr: float):
        self.m = model
        self.layers_per_stage = int(layers_per_stage)
        self.epochs_per_stage = int(epochs_per_stage)
        self.swiftmax_lr = float(swiftmax_lr)
        self.total_layers = len(self.m.bert.encoder.layer)
        self.total_stages = math.ceil(self.total_layers /
          self.layers_per_stage)
        self.stage = 0
        self.epoch_countdown = 0

    def on_epoch_begin(self, args, state, control, model=None,
      optimizer=None, **kwargs):
        if self.stage >= self.total_stages:
            return
        if self.epoch_countdown == 0:
            start = self.stage * self.layers_per_stage
            end   = min(start + self.layers_per_stage, self.total_layers)
            print(f"[SwiftMax] Stage {self.stage+1}/{self.total_stages}:
              "
                  f"replacing layers [{start}..{end-1}]")

            for i in range(start, end):
                layer = self.m.bert.encoder.layer[i]
                old = layer.attention.self
                new = SwiftMaxBertSelfAttention(self.m.config)
                # copy projections & dropout
                new.query, new.key, new.value = old.query, old.key,
                  old.value
                new.dropout = old.dropout
                # swap in
                layer.attention.self = new
                # train beta/gamma
                if optimizer is not None:

                      optimizer.param_groups[0]["params"].extend([new.swiftmax.beta,
                      new.swiftmax.gamma])

            self.stage += 1
            self.epoch_countdown = self.epochs_per_stage

        self.epoch_countdown = max(0, self.epoch_countdown - 1)
```

## B.5  MINIMAL TRAINING ENTRY POINT (GLUE/SST-2 EXAMPLE)

```python
# run_glue_swiftmax_min.py
import math, numpy as np
from datasets import load_dataset
from transformers import (BertForSequenceClassification, BertTokenizer,
                          Trainer, TrainingArguments)
from progressive_callback import ReplaceBertLayersCallback

MODEL = "google-bert/bert-base-uncased"

def compute_metrics(eval_pred):
    preds, labels = eval_pred
    preds = np.argmax(preds, axis=1)
    from evaluate import load
    metric = load("glue", "sst2")
    return metric.compute(predictions=preds, references=labels)

def main(task="sst2",
         layers_per_stage=1, epochs_per_stage=2,
         lr=3e-5, swiftmax_lr=5e-4,
         beta0=10.0, gamma0=2.0, base=math.e):
```

```
        ds = load_dataset("glue", task)
        tok = BertTokenizer.from_pretrained(MODEL)

        def pp(ex):
            if task == "sst2":
                return tok(ex["sentence"], truncation=True,
                ↪ padding="max_length", max_length=128)
            raise NotImplementedError("Only SST-2 shown for brevity.")

        ds = ds.map(pp, batched=True)
        num_labels = ds["train"].features["label"].num_classes
        m = BertForSequenceClassification.from_pretrained(MODEL,
        ↪ num_labels=num_labels)

        # pass SwiftMax hyperparameters via config
        m.config.swiftmax_initial_beta  = float(beta0)
        m.config.swiftmax_initial_gamma = float(gamma0)
        m.config.swiftmax_base          = float(base)

        total_layers = len(m.bert.encoder.layer)
        total_stages = math.ceil(total_layers / layers_per_stage)
        total_epochs = epochs_per_stage * total_stages

        args = TrainingArguments(
            output_dir="./ckpt/bert_sst2_swiftmax",
            evaluation_strategy="epoch",
            save_strategy="epoch",
            learning_rate=lr,
            per_device_train_batch_size=64,
            per_device_eval_batch_size=128,
            num_train_epochs=total_epochs,
            weight_decay=0.01,
            logging_steps=10,
            save_total_limit=1,
            report_to=[],
        )

        cb = ReplaceBertLayersCallback(m, layers_per_stage, epochs_per_stage,
        ↪ swiftmax_lr)
        tr = Trainer(
            model=m,
            args=args,
            train_dataset=ds["train"],
            eval_dataset=ds["validation"],
            tokenizer=tok,
            compute_metrics=compute_metrics,
            callbacks=[cb],
        )
        tr.train()

if __name__ == "__main__":
    main()
```

## B.6 REPRODUCIBILITY NOTES

- **Optimizer and parameters.** In the minimal setup, the newly introduced $\beta, \gamma$ are appended to the main optimizer's first parameter group; this suffices to reproduce the results reported in Section 4.3. If desired, a dedicated parameter group with a higher learning rate can be configured.

- **Numerical stability.** The clamping in §B.2 prevents overflow under mixed precision. We did not require explicit positivity constraints on $\gamma$ given the initialization and optimization settings.

- **Determinism.** For strict reproducibility, fix random seeds and enable deterministic backends (this did not materially change our conclusions).

## C  PRACTICAL WORKFLOW: BASELINE FINE-TUNING → SWIFTMAX REPLACEMENT

**Rationale.**  We first adapt the off-the-shelf pretrained weights to the target dataset via standard fine-tuning, and *then* apply SwiftMax with a lightweight replace-and-tune schedule. This order (pretrained → task-adapted baseline → SwiftMax) yields (i) tighter and more stationary Softmax output statistics for initializing $(\beta, \gamma)$, (ii) higher stability during replacement, and (iii) a fair wall-clock comparison against full fine-tuning.

**Step-by-step.**

1. **Load official pretrained checkpoint** (e.g., BERT-Base or ViT-Base) and the target dataset with standard preprocessing.

2. **Baseline fine-tuning (no SwiftMax).** Train the model on the target task to convergence using the usual recipe (optimizer, LR schedule, epochs). Save the best-validation checkpoint as the *task-adapted baseline*.

3. **(Optional) Collect Softmax statistics.** On a held-out split, record per-layer distributions of $z_{\max}$ and $\sum_j e^{z_j - z_{\max}}$ to initialize $(\beta, \gamma)$ (means or robust percentiles).

4. **Initialize SwiftMax.** Set $\beta_l, \gamma_l$ from the collected statistics (or use the fixed default in App. B.2) and *freeze all original weights*. Train only $\beta_l, \gamma_l$.

5. **Layer-wise replacement and tuning.** Following Alg. 1, progressively replace the Softmax normalizer substep with SwiftMax, running $E$ epochs per stage with a dedicated learning rate for $(\beta, \gamma)$.

6. **Report.** Always report (a) baseline score, (b) SwiftMax score, and (c) wall-clock training time for *baseline fine-tune* vs *SwiftMax replace-and-tune*, on the same hardware.

