# OpenReview forum: "SwiftMax: Reducing Training Time for Learnable Softmax Alternative in Customized Acceleration"
_ICLR.cc/2026/Conference — Submitted to ICLR 2026_

### Official Review · Reviewer_frc8 · 2025-10-28

**Soundness:** 1
**Presentation:** 2
**Contribution:** 1
**Rating:** 2
**Confidence:** 4

**Summary:**

To address the $O(n)$ complexity bottleneck caused by row-wise max and sum operations in the Softmax function of Transformer self-attention, the authors propose SwiftMax, which reduces the complexity of the normalization substep to $O(1)$ by employing two learnable scalars, $\beta$ and $\gamma$, at each layer. This approach significantly decreases the computational cost of Softmax while limiting the accuracy loss to a minimum of 1%.

**Strengths:**

1. Computational and experimental results demonstrate that SwiftMax improves the computation of Softmax while achieving considerable performance gains.​
2. SwiftMax delivers a remarkable 2250x faster training time compared to full retraining. This significant speedup is achieved with a minimal accuracy loss of 1%.

**Weaknesses:**

1. It is well-known that softmax constitutes a relatively small portion of the computation in Transformer-based models. Given that, any improvement focused solely on softmax would inevitably lead to a trade-off: while training efficiency increases, a decline in model accuracy is unavoidable since all other components remain frozen. This trade-off is evidenced by the experimental results, which show a considerable accuracy drop on the RTE task.
2. The existing experiments alone do not prove that SwiftMax improves upon Softmax. In my view, it is essential to compare model accuracy given the same training time. When it's difficult to enhance the optimal accuracy, this at least tells us the specific scenarios where using SwiftMax is more advantageous than using Softmax.
3. The paper lacks critical theoretical derivation. It fails to provide a mathematical proof for the claim that $\beta / \gamma$ can be initialized from pre-trained Softmax statistics and fine-tuned to approximate the original Softmax distribution, relying merely on the experimental observation that the statistics approximate a normal distribution. This results in a weak theoretical foundation. Furthermore, it does not analyze the sensitivity bounds of $\beta / \gamma$—for instance, whether these parameters can still maintain the normalization effect when the input distribution shifts.
4. The paper does not test the impact of sequence length on SwiftMax's effectiveness. While the Softmax bottleneck becomes more pronounced as the sequence length nincreases, the study fails to report accuracy and speedup ratios for long sequences (e.g., n=1024 or 2048), thus leaving the method's efficacy in large-scale sequence scenarios unverified.
5. The core mechanism (using scalars instead of row-wise reduction) is derived from ConSmax. Its innovation lies primarily in optimizing the training strategy rather than introducing a novel normalization mechanism, making it an engineering improvement rather than a theoretical algorithmic breakthrough; thus, it constitutes an incremental contribution. However, I personally believe that, relative to the previous issues, the problem of innovativeness is not even the biggest issue this paper needs to solve.

**Questions:**

Please respond to the concerns I have raised in the 'weaknesses' section. If the revision can adequately address most of the critical issues (although I believe it would be quite challenging), I would consider raising the score.

---

### Official Review · Reviewer_N5as · 2025-10-28

**Soundness:** 2
**Presentation:** 3
**Contribution:** 2
**Rating:** 2
**Confidence:** 4

**Summary:**

This paper proposes SwiftMax, which is based on ConSmax, to address the problem of ConSmax in retraining the model for parameter learning. SwiftMax introduces a fine-tuning strategy that initializes parameters using statistics derived from the pre-trained model and replaces modules layer-by-layer. The method is integrated into BERT-base model and ViT-base model, and evaluated on the GLUE benchmark and the CIFAR-10 dataset, respectively. It is also implemented on AMD ACAP hardware to show the speedup in the self-attention normalizer.

**Strengths:**

1. The idea to reduce the retraining cost in ConSmax is interesting.

2. The implementation of a customizable self-attention block on ACAP is appreciated.

**Weaknesses:**

1. The novelty is limited. The whole algorithm remains almost identical to ConSmax. The work primarily optimizes the fine-tuning pipeline. Hence, there is no improvement in inference speed, which is also critical for acceleration.

2. Figure 2, which seems to be the most important plot, is neither referenced nor explained in the main text.

3. There is no theoretical justification or intuition provided on why the convergence problems occur in the first place and why replacing it layer-by-layer can resolve the issue.

4. The proposed method may not be effective, as the accuracy drops significantly on evaluated tasks. For example, in Figure 4(c), with more layers replaced by SwiftMax, the performance becomes worse. Similar situation happens in the RTE task, where the performance drops by 21.4% (Table 1).

5. Experiments are not comprehensive.

   - The datasets used are relatively small. Large-scale experiments are expected.

   - Only 4 datasets from GLUE Benchmark are evaluated. Why not evaluate on all the datasets from the benchmark? In addition, the SOTA performance in the leaderboard is much higher than the one provided in the text.

   - In Line 379, it is mentioned that "By replacing only a portion of the Self-Attention layers with SwiftMax, we can balance the trade-off between model accuracy and inference speed." However, this is not validated with any experiment.

   - In Section 4.4, the speedup is compared against Online Softmax. How about the accuracy comparison? What is the trade-off here? In addition, evaluation against more recent methods may be needed.

**Questions:**

1. In Figure 4(a) and Figure 4(b), what is the model used here?

2. During the fine-tuning process, is the whole dataset accessible, or only a small portion of the dataset is used?

---

### Official Review · Reviewer_PzqP · 2025-10-31

**Soundness:** 2
**Presentation:** 2
**Contribution:** 2
**Rating:** 2
**Confidence:** 3

**Summary:**

This paper proposes SwiftMax, a practical alternative to the Softmax activation that addresses the computational and memory bottlenecks inherent in standard Softmax operations, especially during the normalization phase requiring sequential reductions.

Building upon the previously introduced ConSmax (Liu et al., 2024), which replaces the Softmax normalization with learnable parameters $\beta$ and $\gamma$ but requires full retraining, the authors propose a method to estimate these parameters directly from pre-trained Softmax output statistics and subsequently refine them through lightweight, layer-wise fine-tuning.

The proposed layer-wise replacement and fine-tuning strategy allows the progressive integration of SwiftMax into pre-trained models, maintaining training stability and numerical robustness. The paper systematically analyzes three key hyperparameters—epochs per stage (E), parameter initialization ($\beta$ and $\gamma$), and learning rate ($\eta$)—to balance convergence stability and efficiency.

Extensive experiments on BERT (GLUE benchmark) and ViT (CIFAR datasets) demonstrate substantial training-time speedups (up to 2250×) with negligible accuracy loss on NLP tasks. The method is further deployed on AMD Adaptive Compute Acceleration Platform (ACAP), achieving up to 23× acceleration in Self-Attention modules compared to traditional and online Softmax implementations.

**Strengths:**

- The paper identifies a real bottleneck in modern neural architectures — the Softmax function’s limited parallelism and memory bandwidth inefficiency. This is a well-known yet under-addressed issue in hardware-efficient deep learning, making the proposed direction highly relevant.

- While ConSmax (Liu et al., 2024) introduced a differentiable alternative to Softmax, it required full retraining to learn parameters $\beta$ and $\gamma$. The proposed SwiftMax overcomes this key limitation by estimating $\beta$ and $\gamma$ directly from Softmax output statistics of pre-trained models, followed by lightweight fine-tuning. This makes SwiftMax far more practical for integration into existing large-scale models.

- The layer-wise replacement and fine-tuning procedure is thoughtfully designed to balance numerical stability and adaptation speed. By progressively introducing SwiftMax in a controlled manner, the method prevents convergence issues common in abrupt global replacements.

- The authors present a detailed empirical analysis of the Softmax output distributions across layers and heads, identifying near-normal characteristics and leveraging them for robust parameter initialization. This empirical grounding supports the plausibility of the proposed estimation process.

- The method is evaluated across multiple models (BERT, ViT) and benchmarks (GLUE, CIFAR) and further validated on AMD’s Adaptive Compute Acceleration Platform (ACAP), achieving up to 23× speedup in Self-Attention modules. Such hardware-level verification adds significant practical value.

**Weaknesses:**

- While SwiftMax performs well on BERT, performance degradation is observed on ViT, especially for CIFAR-100, revealing sensitivity to broader Softmax distributions and architecture-specific dynamics.
- The paper assumes that the Softmax parameter distribution is relatively narrow (as in NLP tasks). However, no mechanism dynamically adjusts $\beta$  and $\gamma$ when faced with broader or non-Gaussian distributions, which limits the method’s general applicability.
- The core hyperparameters—learning rate ($\eta$), epochs per stage (E), layers per stage (L), and initialization of $\beta$ and $\gamma$—are selected empirically. The paper does not provide theoretical reasoning or quantitative sensitivity analysis to demonstrate robustness across models or datasets.
- Different fine-tuning epochs (2 for BERT vs. 3 for ViT) indicate that SwiftMax’s behavior is model-dependent, which undermines its “universal” applicability.
- For system evaluation, a resource-normalized benchmark (e.g., FLOPs, energy, latency) would provide a more rigorous evaluation.
- Fig. 1 and Fig. 2 are cited but not explicitly interpreted or discussed in the text. Please double check.

Overall, SwiftMax is an appealing practical shortcut but lacks theoretical backing, rigorous evaluation, and sufficient breadth of testing to justify its strong claims of efficiency and generality.
The technique seems to work by luck (β, γ approximate normalizer statistics) rather than by provable principle.

**Questions:**

1) Could SwiftMax incorporate an adaptive adjustment for $\beta$ and $\gamma$ during fine-tuning to handle models with broader Softmax output distributions (e.g., ViT, multimodal networks)? Why are $\beta$ and $\gamma$ defined per-layer rather than per-head, given that attention heads often have distinct activation distributions?
2) I do not know if you can consider learning $\beta$ and $\gamma$ as functions of input statistics rather than fixed scalars per layer?
3) How do $\beta$ and $\gamma$ evolve during fine-tuning? Are they stable across epochs or layer depth?
4) How does SwiftMax scale in quantized or mixed-precision settings typical for edge hardware?
5) Since SwiftMax outputs are not normalized to sum to 1, how do you justify that the resulting attention distributions remain meaningful and stable across layers?
6) Beyond freezing pretrained weights, what is the fundamental algorithmic or theoretical difference between SwiftMax and ConSmax? Why is training only $\beta$ and $\gamma$ sufficient?
7) Is there any formal analysis or empirical correlation showing that the learned $\beta$ and $\gamma$ values approximate the true Softmax statistics $z_{max}$ and $\sum_je^{z_j − z_{max}}$?
8) How was the “up to 2,250× faster training” computed? What hardware, batch size, and baseline conditions were used for that figure?
9) How many random seeds or independent runs were used for GLUE and CIFAR experiments, and are the reported accuracy drops statistically significant?
10) How sensitive is performance to the initial $\beta$ and $\gamma$ values or the clamping bounds [−10, 40]? Would a different dataset or sequence length require re-tuning?
11) What happens if only certain attention layers (e.g., shallow vs deep) use SwiftMax? Is there an optimal replacement depth balancing accuracy and speed?
12) The paper reports 23× module-level speedup on ACAP - what is the end-to-end model-level speedup, and how were runtimes measured (simulation vs on-device)?
13) Have you tested SwiftMax on larger or autoregressive transformer architectures (e.g., GPT-style models)? If not, what challenges do you anticipate?

---

### Official Review · Reviewer_5dPR · 2025-11-01

**Soundness:** 3
**Presentation:** 3
**Contribution:** 3
**Rating:** 6
**Confidence:** 3

**Summary:**

The paper presents SwiftMax, a novel, learnable alternative to the Softmax function, designed to mitigate the latency and bandwidth bottlenecks caused by the Softmax. The method replaces the maximum and summation reductions with per-layer learnable scalars $\beta$ and $\gamma$. SwiftMax's key contribution is a method to be applied to pretrained models by fine-tuning only the newly introduced $\beta$ and $\gamma$ parameters. The approach drastically reduces end-to-end training time by up to 2,250$\times$ compared to methods requiring full model retraining.

**Strengths:**

1. In the context of addressing the Softmax bottleneck to drastically improve performance in DNNs, the core strength of the SwiftMax technique is its ability to achieve a constant-time normalizer substep while significantly minimizing the required retraining effort.
2. The approach drastically reduces end-to-end training time by up to 2,250× compared to methods requiring full model retraining.

**Weaknesses:**

1. SwiftMax experiences a significant accuracy drop in some benchmark cases.

**Questions:**

How does the proposed technique compare with the state-of-the-art methods in terms of accuracy? How can the authors address the issue of the accuracy drop, considering that a significant accuracy drop in most applications is unacceptable?

---

### Meta-Review · Area_Chair_1cnf · 2025-12-18

**Summary:**

The paper introduces SwiftMax, a hardware-friendly, learnable alternative to the Softmax function designed to eliminate the $O(n)$ row-wise reduction bottleneck in self-attention. Despite the strong practical motivation, the review process highlighted several fundamental issues with the submission, such as limited technical novelty, significant quality loss, insufficient evaluation, and low-quality presentation. Hence, there is the concensus that the paper is not ready for acceptance in its current form.

**Reviewer Concerns:**

No rebuttal provided.

**Reviewer Scores:**

No rebuttal provided.

---

### Decision · Program_Chairs · 2026-01-26

Reject